

# Genome-wide discovery of circulating cell-free DNA methylation signatures for the differential diagnosis of triple-negative breast cancer

Lijing Gao[1], Yanbing Li[1], Chao Qu[1], Yan Dong[1], Qingzhen Fu[1], Haibo Zhou[1], Ning Zhao[1], Xianyu Zhang[2], Da Pang[2] and Yashuang Zhao[1]

[1] Department of Epidemiology, Harbin Medical University, Harbin, Heilongjiang, China
[2] Department of Breast Surgery, Harbin Medical University, Harbin, Heilongjiang, China

Corresponding authors
Da Pang,
pangda@ems.hrbmu.edu.cn
Yashuang Zhao,
Zhao_yashuang@263.net

## ABSTRACT

**Background:** Preoperative identification of breast cancer (BC) subtypes is essential for optimizing treatment strategies and improving patient outcomes. This study aimed to identify circulating cell-free DNA (cfDNA) methylation signatures to differentiate triple-negative breast cancer (TNBC) from other BC subtypes (non-TNBC).

**Methods:** We initially performed a genome-wide analysis to identify differentially methylated CpG sites (DMCs; $|\Delta\beta| > 0.10$ and $P < 0.05$) between five TNBC and nine non-TNBC tissues using the Infinium HumanMethylationEPIC BeadChip. These DMCs were further validated using large-scale data from the Cancer Genome Atlas (TCGA, $n = 774$; $|\Delta\beta| > 0.25$ and $P < 0.05$), and only CpG sites with average $\beta$ values > 0.90 or < 0.10 in white blood cells (GSE50132, $n = 233$) were retained to minimize potential background methylation interference. Least absolute shrinkage and selection operator (LASSO) regression was applied to select optimal markers. Diagnostic performance was assessed by the area under the receiver operating characteristic curve (AUC), and prognostic value was evaluated using Cox regression analysis. A multiplex digital droplet PCR (mddPCR) assay was developed to simultaneously detect cg06268921 and cg23247845 in cfDNA from TNBC ($n = 33$) and non-TNBC ($n = 80$) patients.

**Results:** We identified 113 DMCs, of which eight were selected as optimal markers. They effectively discriminated TNBC from non-TNBC tissues. Then an eight-marker diagnostic panel was developed with an AUC of 0.922 in TCGA and 0.875 in GSE69914. Among them, cg06268921 was significantly associated with overall survival (hazard ratio = 0.249, $P = 0.044$) and disease-free survival (hazard ratio = 0.194, $P = 0.015$) in the TCGA-TNBC cohort. In the cfDNA cohort, cg06268921 significantly differentiated TNBC from non-TNBC ($P < 0.001$), and the combination of both markers yielded an AUC of 0.728. The findings demonstrated the potential of methylation signatures as non-invasive diagnostic tools for TNBC. Future research with larger cohorts is warranted.

# INTRODUCTION

Triple-negative breast cancer (TNBC) is characterized by the absence of estrogen receptor, progesterone receptor, and human epidermal growth factor receptor 2 expression, accounting for 15% to 20% of all BC cases (*Foulkes, Smith & Reis-Filho, 2010*; *Perou, 2011*). TNBC is considered the most aggressive subtype due to rapid growth, early relapse, and frequent metastasis (*Bianchini et al., 2016*; *Ensenyat-Mendez et al., 2023*; *Metzger-Filho et al., 2012*), leading to poor survival compared to other subtypes of BC (*Qiu et al., 2016*). It usually responds poorly to endocrine therapy and human epidermal growth factor receptor 2 (HER2)-targeted therapy (*Yagata, Kajiura & Yamauchi, 2011*), so if TNBC and other BC subtypes (non-TNBC) are distinguishing before surgery, it can been early determined whether patients are suitable for neoadjuvant treatment and which strategy would be more effective and also plays an important role in improving survival (*Zhang et al., 2021a*). However, the current differential diagnosis of TNBC primarily relies on immunohistochemistry, which involves in highly invasive tissue biopsy sampling and can be time-consuming (*Bianchini et al., 2016*; *Dass et al., 2021*). Mammography and ultrasound have been used for screening and diagnosis of BC overall (*Wang et al., 2022*; *Xi et al., 2022*), until recently these imaging methods were studied to distinguish TNBC from non-TNBC, but with result of relatively low discriminating power for microtumor and largely limited by the experience of the radiologist (*Ma et al., 2022*; *Shaikh & Rasheed, 2021*). In addition, these imaging methods are often limited by the discomfort of patients, radiation exposure, or high false-positive rates issues (*Lee et al., 2010*; *Vourtsis & Berg, 2019*). Consequently, a non-invasive, safer, and more accurate approach needs to be developed to aid in differential diagnosis of TNBC.

DNA methylation is a heritable alteration that modulates gene expression without any change in DNA sequence (*Bird, 2002*). Aberrant DNA methylation changes occur in early carcinogenesis and hold potential as diagnostic markers for cancers, including BC (*Baylin & Jones, 2016*; *Vietri et al., 2021*). In a liquid biopsy, circulating cell-free DNA (cfDNA) is emerging as a non-invasive marker for early cancer diagnosis (*Mattox et al., 2019*; *Schrag et al., 2023*; *Zhang et al., 2024*). Several studies have evaluated the usefulness of the tumor methylation patterns of cfDNA in BC diagnosis. For instance, *Manoochehri et al. (2023)* found that cfDNA-based methylation could discriminate TNBC from healthy controls with an area under the receiver operating characteristic curve (AUC) of 0.780 in the test set and 0.740 in the validation set. In addition, researchers also found that cfDNA methylation patterns could detect BC from benign tumors (*Liu et al., 2021*). Although these studies demonstrated potential in distinguishing cancers from healthy controls or benign tumors, they did not specifically address the differential diagnosis between the subtypes of BC. Hence, the methylation markers that could distinguish TNBC from non-TNBC are lacking and DNA methylation-based prognostic stratification of TNBC is rarely studied.

Therefore, in our study, we initially identified significantly differentially methylated CpG sites (DMCs) between TNBC and non-TNBC tissues using the Infinium HumanMethylationEPIC BeadChip. Then these DMCs were validated by the Cancer Genome Atlas (TCGA) and Gene Expression Omnibus (GEO) datasets. Subsequently, we applied the least absolute shrinkage and selection operator (LASSO) analysis to select the optimal methylation markers. Furthermore, we assessed the diagnostic and prognostic value of these markers in tissues. Finally, two candidate markers were detected in a cfDNA validation cohort using a multiplex digital droplet PCR (mddPCR) assay, demonstrating their potential to distinguish TNBC from non-TNBC.

## MATERIALS AND METHODS

### Patients and sample collection

This study recruited 113 primary BC patients diagnosed by pathology at Cancer Hospital of Harbin Medical University between May 2021 and December 2023. The exclusion criteria included: (1) incomplete molecular subtype data; (2) prior diagnosis of other malignancies; (3) any therapy received before surgery. Tumor tissues ($n = 14$; 5 TNBC and 9 non-TNBC) were obtained through surgical resection and immediately snap-frozen in liquid nitrogen. These samples were used for genome-wide DNA methylation profiling using the Infinium HumanMethylation850K BeadChip to identify candidate markers. Blood samples (1–2 mL, $n = 113$) were collected using Ardent Cell-Free DNA blood tubes and centrifuged at 800 g at room temperature for 10 min to separate plasma and buffy coat. Plasma was then transferred to a new tube and centrifuged at 16,000 g at 4 °C for 10 min to remove the remaining cells. Plasma and buffy coat were immediately stored at −80 °C until analysis. cfDNA extracted from the plasma of all enrolled patients (33 TNBC and 80 non-TNBC) was used to validate selected methylation markers through mddPCR assay. Demographic and clinical data were collected from medical records of all patients. This study was approved by the Medical Ethics Committee of Harbin Medical University (KY2016-01). Written informed consent was obtained from all patients before enrollment and sample collection.

### Public data sources

The Illumina-normalized methylation data of tissues from the TCGA BC cohort, including 83 TNBC and 691 non-TNBC, were downloaded from UCSC Xena (https://xena.ucsc.edu/public/). Additionally, Illumina-normalized methylation data from 342 BC tissues and 233 BC white blood cells (WBC) were downloaded from GEO database (GSE69914, GSE72251, and GSE50132). The methylation level for each CpG was represented as a beta value ($\beta$), which is a ratio of intensities between the methylated allele and the sum of M and unmethylated, ranging from 0 (no methylation) to 1 (full methylation).

### DNA extraction and bisulfite conversion

Tumor tissue DNA was extracted using the QIAamp DNA Mini Kit (Qiagen, Hilden, Germany) and quantified using NanoDrop 2000 (Thermo Fisher Scientific, Waltham, MA, USA). Circulating cfDNA was extracted from plasma using the QIAamp® Circulating
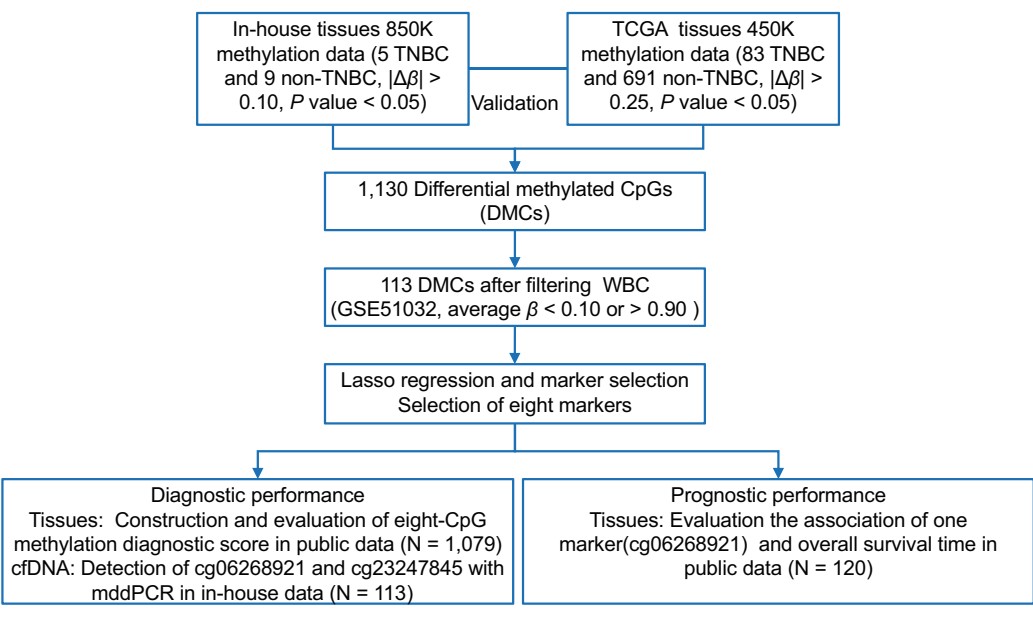

**Figure 1** **The overall flowchart of the study.** Different $\Delta\beta$ thresholds were applied considering sample size differences between datasets (in-house: $|\Delta\beta| > 0.10$; TCGA: $|\Delta\beta| > 0.25$). TNBC, Triple-negative breast cancer; CpG sites; DMCs, differentially methylated CpG sites; WBC, white blood cells; Lasso, least absolute shrinkage and selection operator; mddPCR, multiplex digital droplet PCR; cfDNA, cell-free DNA.

Nucleic Acid Kit (Qiagen, Hilden, Germany) according to the manufacturer's instructions, with concentrations measured using a Qubit 3.0 fluorometer (Thermo Fisher Scientific, Waltham, MA, USA). Both cfDNA and genomic DNA underwent bisulfite conversion using the EZ DNA Methylation-Gold™ Kit (Zymo Research, Irvine, CA, USA), according to the manufacturer's guidelines.

## Tissue methylation markers discovery and selection

As shown in Fig. 1, we first performed the genome-wide methylation analysis on five in-house TNBC and nine non-TNBC tissues (Table S1) using Infinium MethylationEPIC BeadChip. After data filtering, correction, and normalization, CpG sites with $|\Delta\beta| > 0.10$ and $P < 0.05$ were identified as DMCs. Given the limited sample size of the in-house dataset, a more lenient threshold was used to include more potential candidate markers. To improve reliability, these markers were further evaluated in a larger cohort of 83 TNBC and 691 non-TNBC tissues from the TCGA dataset, using a stricter cutoff ($|\Delta\beta| > 0.25$ and $P < 0.05$). Subsequently, to minimize false positive detection and increase the signal-to-noise ratio in cfDNA analysis, CpG sites that were almost completely methylated (average $\beta$ value $> 0.90$) or unmethylated (average $\beta$ value $< 0.10$) in WBC were selected. Pearson's correlation method was used to calculate correlation for methylation markers. Finally, considering the existence of multicollinearity among them, the LASSO analysis with 10-fold cross-validation was performed to select the optimal markers through the "glmnet" package (*Friedman, Hastie & Tibshirani, 2010*) in the TCGA dataset. The eight tissue methylation markers with non-zero coefficients were selected for further analysis.

## Evaluation and validation the diagnostic effect of DNA methylation markers in tissues

The diagnostic accuracy of individual CpG sites in differentiating TNBC and non-TNBC tissues was assessed using ROC curves. CpG sites with an AUC greater than 0.80 were retained. An eight-CpG methylation diagnostic score (MDS) was then constructed using logistic regression in TCGA dataset and validated in GSE69914 dataset. It was calculated based on the corresponding coefficients of the selected methylation markers. The formula was as follows:

$$\text{MDS} = \sum_{i=1}^{n}(m_i * Coef_i + b).$$

Here, $n$ represents the number of methylation markers; $m_i$ represents the methylation level of each marker; $Coef_i$ represents the coefficients; $b$ represents to the intercept; and MDS represents a weighted sum of the methylation level of each marker.

## Evaluation and validation the prognostic effect of DNA methylation markers in TNBC tissues

The primary endpoint for prognostic analysis in the study was overall survival (OS), defined as the time from surgery to death from any cause or the last follow-up visit. In addition, disease-free survival (DFS) was evaluated as a secondary endpoint, defined as the time from surgery to the first local recurrence, distant metastasis, or death from any cause. For each candidate CpG site, patients were divided into low-risk and high-risk groups based on median methylation levels. Univariate Cox regression analysis was performed to screen out CpG sites that significantly associated with OS or DFS in the TCGA-TNBC patients ($P < 0.10$). Multivariate Cox regression was subsequently performed, incorporating the significant CpG sites as predictor variables and adjusting for age and stage, to identify independent prognostic markers. The stability of these markers was validated in the GSE72251-TNBC cohort.

## Development of mddPCR assay

We successfully designed forward and reverse primers, as along with minor groove binder (MGB) Taqman probes (FAM™/VIC™-reporter dyes) for two of the targeted markers (Table S2), while design attempts for the others were unsuccessful due to unfavorable sequence features. The methylation-insensitive β-actin (ACTB) gene served as an internal control in each PCR well (primers and probes reported elsewhere) (Zhang et al., 2023). A mddPCR assay (including three genes) was then performed in a 21 µL final volume system with 10 µL of ddPCR™ Supermix for Probes (No dUTP), adjusted volumes of primers and probes, and 5–6 µL bisulfite-converted DNA. The QX200™ Droplet Generator (Bio-Rad, Hercules, CA, USA) generated droplets and the PCR conditions were as follows: 10 min at 95 °C, 40 cycles of 94 °C for 30 s, and annealing and extension at 60 °C for 1 min; 10 min hold at 98 °C. The ramp rate was set at 2 °C/s for all steps. The emulsions were analyzed on the QX200™ Droplet Reader device to count droplets containing amplified DNA targets and empty droplets based on fluorescence. Data were analyzed using the QuantaSoft™

Analysis Pro software v1.0. Positive droplets (containing an amplified DNA target) were manually identified in a 2D plot based on the distribution of droplets of control samples (methylated DNA control, unmethylated DNA control, RNase-free water control, and non-template control). cfDNA methylation was quantified as the number of copies of methylated alleles per 1 mL of plasma.

## Statistical analysis

Categorical variables were described as the number (percentage), and continuous variables as median (interquartile range (IQR)) or mean (standard deviation (SD)). Methylation levels between the two groups were compared using the $t$-test, Wilcoxon test, or Mann-Whitney U test. Differential methylation analysis between TNBC and non-TNBC tissues were conducted using ChAMP (2.32.0) package (*Tian et al., 2017*) or "Limma" package (*Ritchie et al., 2015*). The unsupervised hierarchical clustering heatmap of DMCs methylation was generated using the "pheatmap" package (*Kolde, 2019*). Gene Ontology (GO) and Kyoto Encyclopedia of Genes and Genomes (KEGG) pathway enrichment analyses (*Gene Ontology Consortium, 2021*; *Kanehisa et al., 2021*) were conducted using the "clusterProfiler" package (*Wu et al., 2021*). Sensitivity, specificity, and AUC were used to evaluate the diagnostic performance of the individual CpG sites and models. The R package "survival" (*Therneau, 2024*) and "survminer" (*Kassambara, Kosinski & Biecek, 2021*) were used for survival analysis. The Kaplan-Meier and log-rank tests were utilized to generate the survival curves and compare the differences between the groups. Time-dependent ROC curves were performed to evaluate the prognostic performance of CpG sites by the R package "timeROC" (*Blanche, Dartigues & Jacqmin-Gadda, 2013*). Hazard ratios (HR) and 95% confidence interval (CI) were calculated with the Cox regression analysis. A minimum sample size of 31 in each group could achieve 80% power with a significance level of 0.05 using a one-sided z-test, which was conducted with PASS 11.0 (UCSS, USA) (*Hanley & McNeil, 1983*; *Obuchowski & McClish, 1997*). Statistical analyses were performed in R (version 4.3.1), and $P < 0.05$ was considered significant.

# RESULTS

## Basic characteristics of patients

A total of 33 TNBC and 80 non-TNBC patients were enrolled in this study. The age distribution was relatively balanced between TNBC and non-TNBC groups (mean age, 59.5 years $vs.$ 58.7 years, $P = 0.800$). Detailed characteristics of patients are summarized in Table 1 and Table S3.

## Differential methylation markers based on in-house primary tissues

Based on a genome-wide methylation analysis between in-house five TNBC and nine non-TNBC tissues using Infinium HumanMethylationEPIC BeadChip, a total of 32,787 DMCs ($|\Delta\beta| > 0.10$ and $P < 0.05$) were identified; with 9,294 (28.35%) DMCs higher methylation levels in TNBC tissues (defined as hypermethylated CpG) and 23,493 (71.65%) with lower methylation levels (defined as hypomethylated CpG, Fig. 2A). An overview of the relative CpG island position and gene location annotation were shown in

**Table 1 The basic characteristics of patients in in-house cohort.**

| Characteristic | Non-TNBC (N = 80) | TNBC (N = 33) | P value |
|---|---|---|---|
| Age (years) | | | 0.800 |
| Mean ± SD | 59.5 ± 10.5 | 58.7 ± 9.2 | |
| Range | 36–82 | 37–73 | |
| BI-RADS category in mammography[a], n (%) | | | 0.032 |
| 1-4a | 22 (27.5%) | 11 (33.3%) | |
| 4b-6 | 48 (60.0%) | 12 (36.4%) | |
| Unknown | 10 (12.5%) | 10 (30.3%) | |
| BI-RADS category in ultrasound[a], n (%) | | | 0.150 |
| 1-4a | 5 (6.3%) | 4 (12.1%) | |
| 4b-6 | 61 (76.2%) | 19 (57.6%) | |
| Unknown | 14 (17.5%) | 10 (30.3%) | |
| Tumor size, n (%) | | | 0.300 |
| Tumor size <= 2 cm | 38 (47.5%) | 19 (57.6%) | |
| Tumor size > 2 cm | 42 (52.5%) | 14 (42.4%) | |
| Lymph node, n (%) | | | 0.400 |
| Negative | 54 (67.5%) | 19 (57.6%) | |
| Positive | 26 (32.5%) | 14 (42.4%) | |
| Unknown | 0 (0.0%) | 0 (0.0%) | |
| Stage[b], n (%) | | | 0.300 |
| Stage I | 29 (36.3%) | 15 (45.5%) | |
| Stage II | 34 (42.5%) | 11 (33.3%) | |
| Stage III | 16 (20.0%) | 5 (15.2%) | |
| Stage IV | 1 (1.3%) | 2 (6.1%) | |
| CEA, n (%) | | | 0.130 |
| ≤5 ng/mL | 64 (80.0%) | 25 (75.8%) | |
| >5 ng/mL | 6 (7.5%) | 0 (0.0%) | |
| Unknown | 10 (12.5%) | 8 (24.2%) | |
| CA15-3, n (%) | | | 0.130 |
| ≤25 U/mL | 65 (81.3%) | 25 (75.8%) | |
| >25 U/mL | 5 (6.3%) | 0 (0.0%) | |
| Unknown | 10 (12.5%) | 8 (24.2%) | |

Notes:
[a] Categories of the Breast Imaging Reporting and Data System.
[b] Clinical staging was determined according to the eighth edition of the classification for breast cancer of the American Joint Commission of Cancer;
SD, Standard deviation; TNBC, Triple-negative breast cancer; CEA, Carcinoembryonic antigen; CA15-3, Cancer antigen 15-3.

the Sankey plot (Fig. 2B). Pathway enrichment analysis showed that the DMCs were mainly enriched in signaling-related pathways (Fig. 2C) and the biological process of protein binding (Fig. S1).

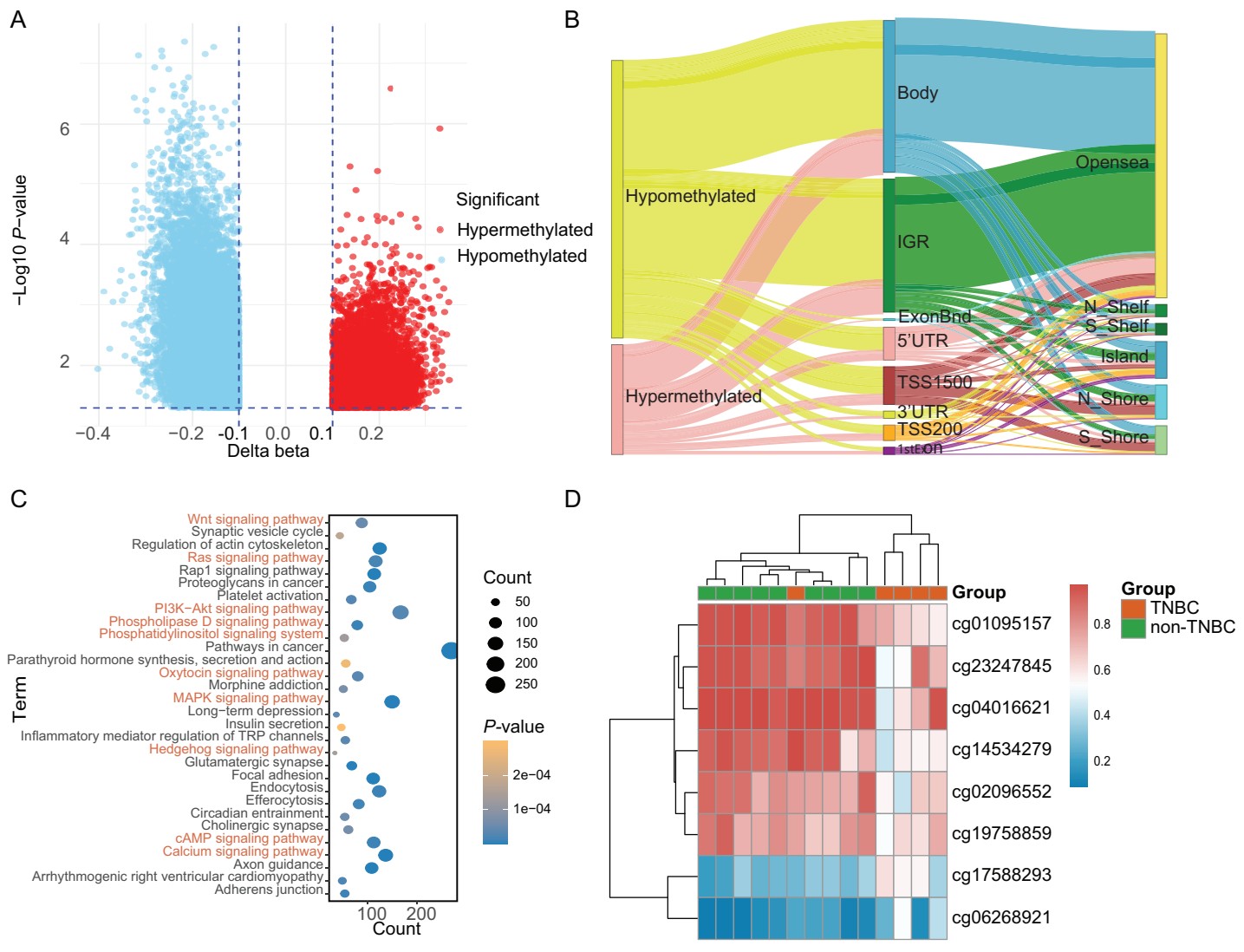

**Figure 2 Identification of differentially methylated CpG sites between TNBC and non-TNBC tissues based on in-house 850K data.**
(A) Volcano plot illustrating the hypermethylated- and hypomethylated-DMCs. (B) Sankey plot of the gene location relative to their genomic distribution of DMCs. (C) The Kyoto Encyclopedia of Genes and Genomes enrichment analysis of differentially methylated genes. (D) Heatmap of eight DMCs between TNBC and non-TNBC tissues. DMCs, differentially methylated CpG sites; TNBC, Triple-negative breast cancer.

## Discovery and selection of tissue-based methylation markers

Based on above obtained 32,787 DMCs from the in-house dataset, the methylation data of 83 TNBC and 691 non-TNBC tissues from TCGA were compared and selected 1,130 CpG that were still significantly differentially methylated ($|\Delta\beta| > 0.25$ and $P < 0.05$). Since cfDNA can be derived from WBC of cancer patients, we excluded the CpG sites with average methylation $\beta \geq 0.10$ or $\beta \leq 0.90$ in 233 WBC to reduce the false positive possibility, and we obtained 113 DMCs. The 113 methylation markers showed high correlations among them (Fig. S2). To further determine CpG sites that could distinguish TNBC from non-TNBC, we applied the Lasso analysis to select optimal markers in TCGA

**Table 2 The genomic characteristics of methylation CpG sites.**

| CpG sites | Gene | Gene description | Genomic coordinate | Gene_Group | CpG_Island |
|---|---|---|---|---|---|
| cg01095157 | *GORASP2* | Golgi Reassembly Stacking Protein 2 | chr2:171784674 | TSS1500 | N_Shore |
| cg02096552 | *DISP1* | Dispatched RND Transporter Family Member 1 | chr1:223168232 | Body | Opensea |
| cg04016621 | *GRK7* | G Protein-Coupled Receptor Kinase 7 | chr3:141495947 | TSS1500 | N_Shore |
| cg06268921 | *NA* | | chr1:214158573 | IGR | N_Shore |
| cg14534279 | *NA* | | chr10:3329966 | IGR | Opensea |
| cg17588293 | *ZBTB7B* | Zinc Finger and BTB Domain Containing 7B | chr1:154986238 | 5′UTR | N_Shelf |
| cg19758859 | *SASH1* | SAM and SH3 Domain Containing 1 | chr6:148869489 | Body | Opensea |
| cg23247845 | *NA* | | chr10:3679085 | IGR | Opensea |

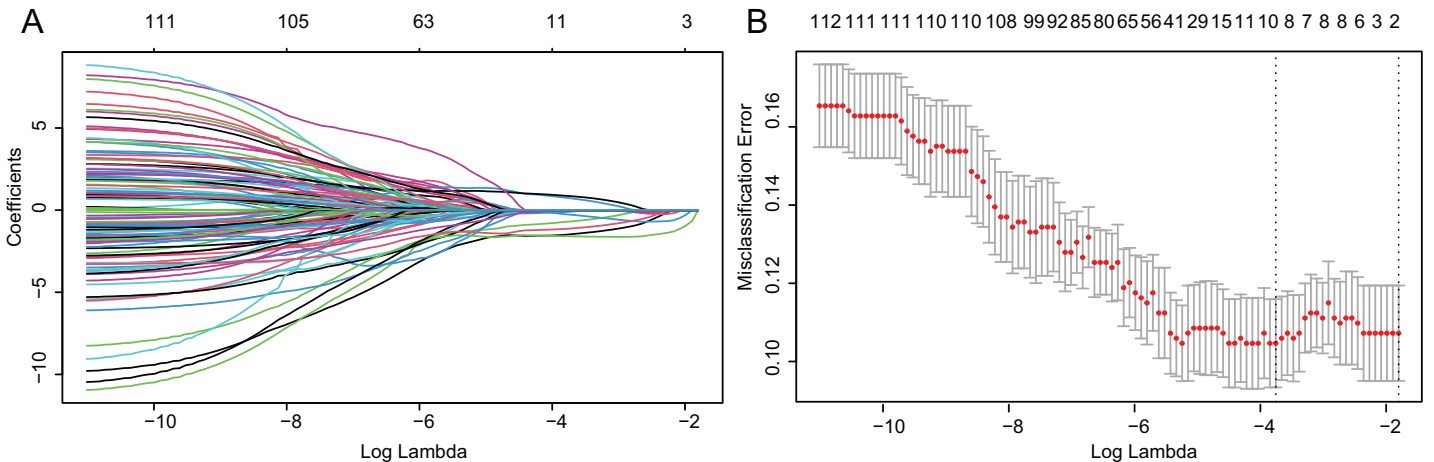

**Figure 3 Identification of optimal methylation markers for distinguishing TNBC from non-TNBC tissues.** (A) Selection of optimal markers in the LASSO model. (B) Tenfold cross-validation for tuning parameter (lambda) selection in the LASSO model. The dotted vertical lines were drawn at the optimal values using the maximum criteria and the one standard error of the maximum criteria. LASSO, least absolute shrinkage and selection operator; TNBC, Triple-negative breast cancer.

dataset. We obtained eight markers (cg19758859, cg01095157, cg14534279, cg17588293, cg06268921, cg04016621, cg23247845, and cg02096552) with non-zero coefficients for further analysis, and the optimal λ value = 0.0234, log (λ) = −3.7547 (Table 2 and Fig. 3). The methylation levels for two CpG sites were significantly higher in TNBC than those in non-TNBC tissues, six residual CpG sites were significantly lower in TNBC than those in non-TNBC tissues (Fig. 2D and Table S4).

## Diagnostic performance of candidate markers in tissues

We investigated the diagnostic performance of eight CpG sites, with AUCs ranging from 0.809 to 0.893 for differentiating TNBC from non-TNBC tissues (Table S5). These CpG sites (AUC > 0.80) were used to construct the MDS through logistic regression analysis. The MDS was a sum of the methylation levels of eight CpG sites, each multiplied by its corresponding coefficient from the logistic regression analysis: MDS = cg19758859 × (−0.394) + cg01095157 × (−1.894) + cg14534279 × (−1.088) + cg17588293 × 1.317 +

cg06268921 × 1.207 + cg04016621 × 0.126 + cg23247845 × (−1.559) + cg02096552 × (−1.489) + 1.189. The MDS was significantly different between TNBC and non-TNBC tissues. (Fig. 4A), with an AUC of 0.922 (95% CI = 0.895–0.950) in the TCGA dataset (Fig. 4B). Using the cutoff value of −2.51, the MDS yielded a sensitivity of 93.8% (61/65) in stage I/II, 82.4% (14/17) in stage III/IV at a specificity of 84.9% (587/691, Table S6). Furthermore, there was no significant difference in MDS between stage I/II and stage III/IV TNBC tissues (Fig. S3A). Notably, the MDS could discriminate the early-stage TNBC from non-TNBC, with an AUC of 0.932 (95% CI [0.908–0.957], Fig. S3B).

The MDS was then applied to the GSE69914 validation dataset, where it also showed a significant difference between TNBC and non-TNBC tissues (Fig. 4C). Using the same cutoff value of −2.51, the MDS achieved an AUC of 0.875 (95% CI [0.789–0.961], Fig. 4D), with a sensitivity of 86.70% (26/30) and a specificity of 90.2% (248/275).

## Prognostic effect of candidate markers in tissues

Next, we investigated prognostic potential of the eight CpG sites in the TCGA-TNBC cohort by analyzing their association with OS and DFS. Among them, only cg06268921 (≥0.50 *vs.* <0.50) was significantly associated with both OS and DFS in univariate Cox regression analysis ($P < 0.10$; Tables S7 and S9). After adjusting for age and stage in multivariate Cox regression analysis, cg06268921 remained significantly associated with OS (HR: 0.249, 95% CI [0.064–0.966], $P = 0.044$) and DFS (HR: 0.194, 95% CI [0.052–0.727], $P = 0.015$), suggesting that it served as an independent prognostic factor for TNBC (Tables S8 and S10). The Kaplan–Meier survival curves further confirmed that patients in the low-methylation group (<0.50) had significantly worse OS and DFS compared with those in the high-methylation group (≥0.50; log-rank $P < 0.10$, Figs. 5A, 5B). The prognostic performance of cg06268921 was evaluated using time-dependent ROC analysis, yielding AUCs of 0.521 and 0.422 for 3-year OS and DFS, respectively (Figs. S4A, S4B). To validate our findings, we analyzed the prognostic effect of cg06268921 in an independent external cohort (GSE72251). The association was not statistically significant, possibly due to limited sample size and cohort heterogeneity (Tables S8 and S10). Meta-analysis of both cohorts showed non-significant pooled HR for OS and DFS, with moderate heterogeneity ($I^2 > 60\%$), indicating variability in the prognostic effect of cg06268921 across cohorts (Fig. S5).

## Development of cfDNA mddPCR assay and the evaluation of limit of quantification

We constructed an mddPCR assay (cg06268921, cg23247845, and *ACTB*, Fig. 6A and Table S11) and then optimized its annealing temperature. By comparing the fluorescent of methylated and unmethylated control in the FAM channel, we found that an annealing temperature of 58.8 °C yielded the best performance (Fig. S6).

Next, we compared the limit of quantification (LOQ) of mddPCR assay with multiplex quantitative methylation-specific PCR (mqMSP) assay using 10 ng of mixed methylated DNA control and the unmethylated DNA control at concentrations of methylation at 100%, 10%, 5%, 1%, 0.5%, 0.1%, 0.01%, and 0. Three replicates of each reaction were

 

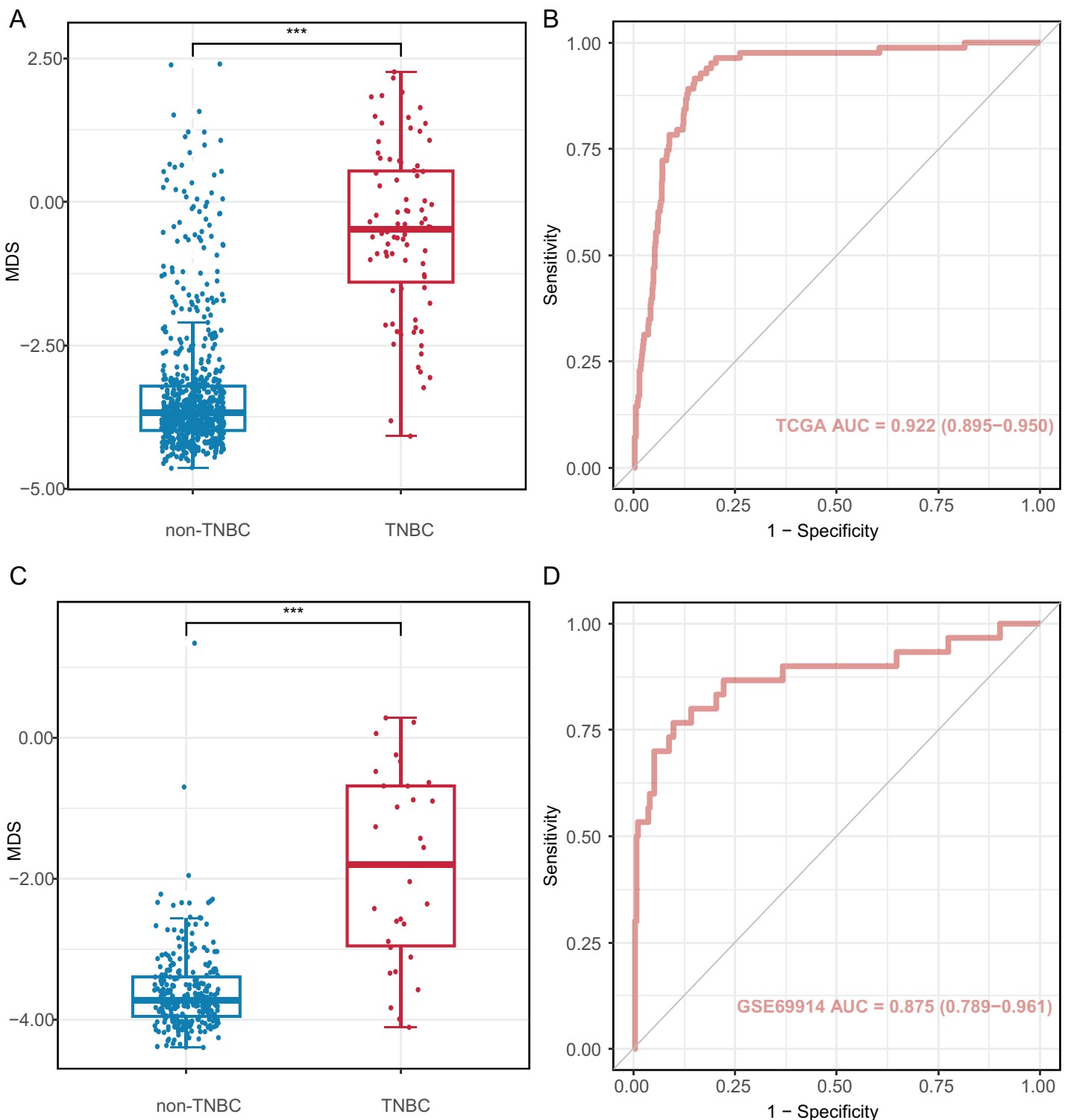

**Figure 4 Diagnostic performance of methylation diagnostic score (MDS) in tissues.** (A and C) Boxplot and dotplot of MDS for TNBC and non-TNBC in the TCGA and GSE69914 datasets. (B and D) Receiver operating characteristic curve of MDS for distinguishing TNBC and non--TNBC in the TCGA and GSE69914 datasets. TNBC, Triple-negative breast cancer. "***" means $P \leq 0.001$.

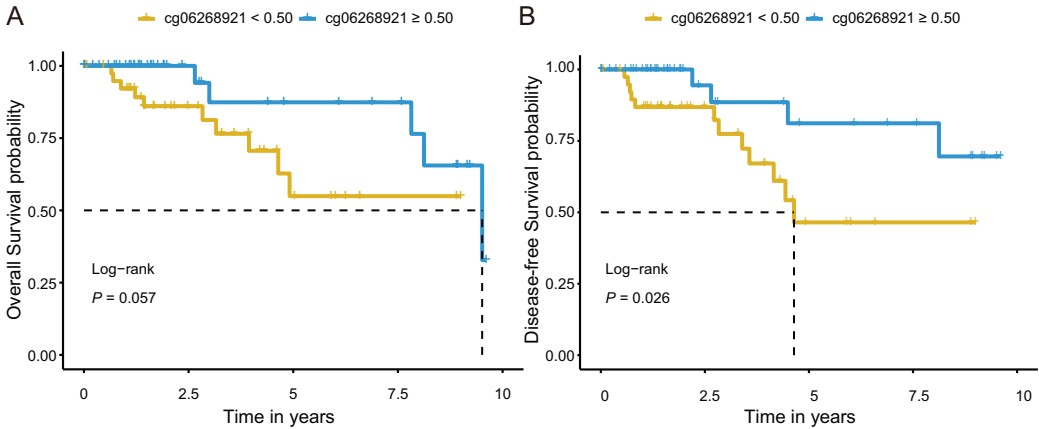

**Figure 5 Kaplan–Meier survival analysis of cg06268921 methylation levels in TNBC patients from the TCGA cohort.** (A) Overall survival (OS) curve stratified by cg06268921 methylation level (cut-off = 0.50); (B) Disease-free survival (DFS) curve stratified by cg06268921 methylation level (cut-off = 0.50); Patients in the low-methylation group exhibited significantly worse OS and DFS compared to those in the high-methylation group (Log-rank $P < 0.10$). TNBC, Triple-negative breast cancer; TCGA, the Cancer Genome Atlas.                     

performed. The mddPCR assay could simultaneously and independently quantify the methylation level of each target gene, while mqMSP could only determine the combined methylation level of multiple genes. In the FAM channel of the mqMSP assay, one methylated allele in a background of 20 unmethylated alleles was detected (LOQ = 5%, $R^2 = 0.982$). In contrast, the LOQ of cg06268921 ($R^2 = 0.790$) and cg23247845 ($R^2 = 0.990$) of mddPCR were 0.01% and 0.1%, respectively, indicating that the LOQ of mddPCR assay was 0.01%, and greater sensitivity than the mqMSP assay (Fig. S7).

## Validation the diagnostic effect of methylated markers detected with mddPCR in cfDNA cohort

Moreover, we applied the mddPCR assay to detect methylation levels in cfDNA from 33 TNBC and 80 non-TNBC patients. The TNBC and non-TNBC yielded the median cfDNA concentration of 7.15 ng/mL (IQR: 4.58–9.80), and 6.98 ng/mL (IQR: 2.54–10.4), respectively, with no statistically significant different ($P = 0.97$, Fig. S8). However, the number of methylated molecular copies of cg06268921 was significantly higher in TNBC than in non-TNBC subtypes with AUC of 0.719 (95% CI [0.614–0.823]), but cg23247845 was not significantly different between two groups (Fig. S9). We incorporated these two markers to construct the cfDNA-based MDS (cf-MDS), which was significantly different between TNBC and non-TNBC groups in cfDNA and achieved a sensitivity of 54.5%, a specificity of 82.5%, and an AUC of 0.728 (95% CI [0.618–0.839], Figs. 6B, 6C). In addition, the cf-MDS could distinguish stage I/II TNBC from non-TNBC, with an AUC of 0.701 (95% CI [0.578–0.825], Fig. S10).

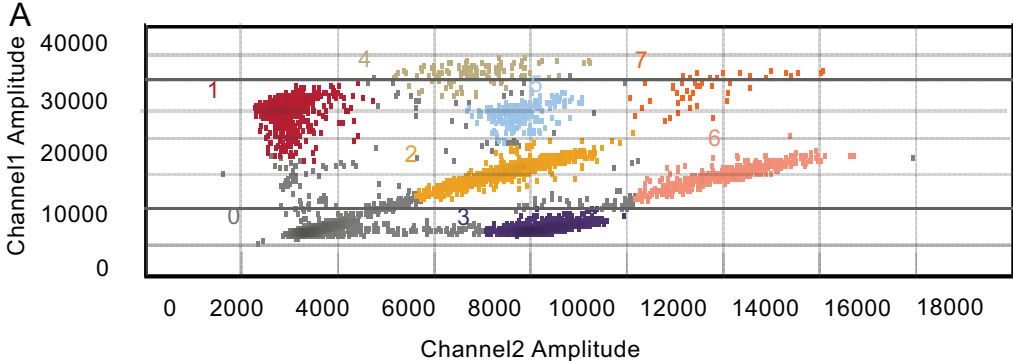

0: empty; 1: cg06268921; 2: cg23247845; 3: *ACTB*; 4: cg06268921+cg23247845;
5: cg06268921+*ACTB*; 6: cg23247845+*ACTB*; 7: cg06268921+ cg23247845+*ACTB*

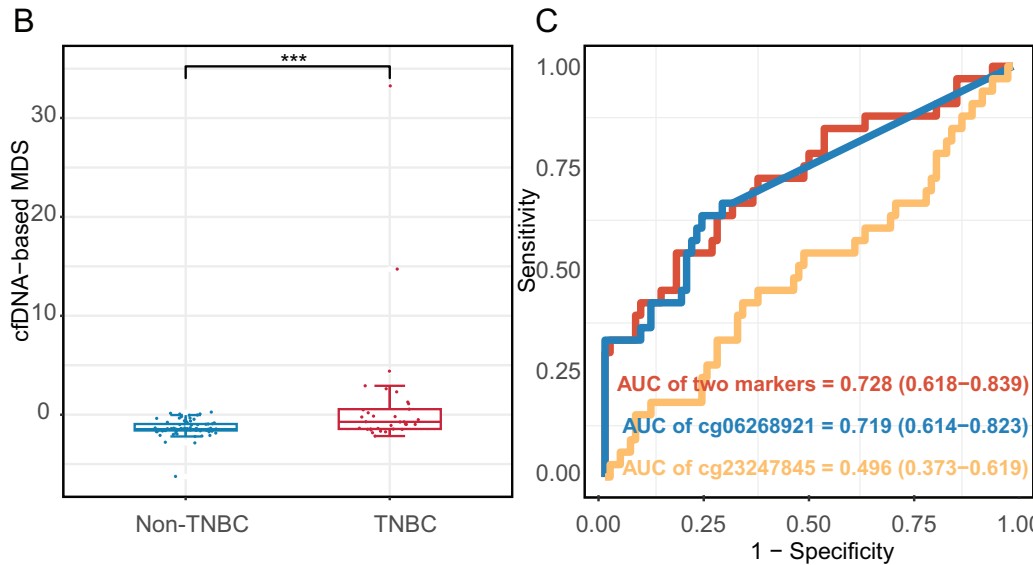

**Figure 6 Diagnostic effect of methylation markers in plasma cell-free DNA.** (A) Analysis of mddPCR assay for cluster definitions, including empty droplets, cg06268921, cg23247845, and *ACTB*, and double-positive or triple-positive droplet clusters; Channel 1 amplitude: FAM fluorescence; Channel 2 amplitude: VIC fluorescence. (B) Boxplot and dotplot of the cfDNA-based MDS for TNBC and non-TNBC in plasma cell-free DNA. (C) Receiver operating characteristic curve of methylation markers for distinguishing TNBC from non-TNBC in cfDNA. ***: $P \leq 0.001$; TNBC, Triple-negative breast cancer; cfDNA, cell-free DNA.

## DISCUSSION

In this study, we discovered cfDNA methylation markers for the differential diagnosis of TNBC. Firstly, we identified 113 DMCs based on in-house tissue Infinium HumanMethylationEPIC dataset combined with TCGA and GEO datasets. Subsequently, we employed Lasso regression analysis to further select the eight optimal methylation markers. The eight-CpG MDS could distinguish TNBC from non-TNBC tissues, with AUC over 0.850. The Cox regression analysis revealed that cg06268921 was significantly associated with OS and DFS in TCGA-TNBC cohort. Furthermore, in our cfDNA validation cohort, we developed a mddPCR assay to detect two of these markers, the

cf-MDS has potential as a non-invasive diagnostic marker for differentiating TNBC from non-TNBC.

DNA methylation is a common early epigenetic modification in tumorigenesis and can be tissue-specific, making it a valuable foundation for cancer diagnostic markers, including those for BC (*Kanwal, Gupta & Gupta, 2015*; *Laird, 2003*; *Liu et al., 2021*; *Zhang et al., 2021b*). Several studies have also demonstrated significant differences in DNA methylation profiles across BC subtypes. For instance, *Sunami et al. (2008)* reported distinct methylation patterns of genes such as *RASSF1A*, *CCND2*, *GSTP1*, *TWIST*, and *APC* between the estrogen receptor (ER)-positive and ER-negative groups. Similarly, *Fackler et al. (2011)* identified 40 CpG loci capable of distinguishing ER-positive from ER-negative BC, achieving an AUC of 0.961 in TCGA data, underscoring the robustness of DNA methylation as a subtype-specific marker. The existing studies have focused comparisons between TNBC and normal or tumor-adjacent tissues, or the development of prognostic markers (*Cristall et al., 2021*; *DiNome et al., 2019*; *Lin et al., 2023*; *Manoochehri et al., 2023*). However, few studies have specially profiled the DNA methylation profile between TNBC and non-TNBC. In our study, we first conducted genome-wide DNA methylation profiling to compare TNBC and non-TNBC tissues, identifying a set of DMCs. Functional enrichment analysis revealed that these DMCs were predominantly involved in key signaling pathways, including the Wingless/Integrated (Wnt), Phosphatidylinositol 3-kinase/Protein kinase B (PI3K-Akt), and Mitogen-Activated Protein Kinase (MAPK) signaling pathways, which are known to regulate tumor cell proliferation, invasion, and metastasis (*Chen, Zhang & Dai, 2019*). Additionally, the enrichment in protein binding functions suggests potential epigenetic regulation of key transcriptional in cancer progression. Building upon these findings, we further selected eight TNBC-specific methylation markers with strong discriminatory power to construct MDS. The MDS demonstrated superior sensitivity compared to individual markers, with an AUC of 0.922, highlighting its potential as a robust tool for distinguishing TNBC from non-TNBC. Notably, the panel of 282 methylation markers proposed by *Stirzaker, Zotenko & Clark (2016)* achieved a sensitivity of 0.720 (AUC = 0.900) for distinguishing TNBC from non-TNBC tissues in TCGA data, which is lower than that observed in our study using TCGA data (Sensitivity = 0.916). We speculated that the lower sensitivity may be due to their markers being identified through comparisons between TNBC and matched normal tissues, rather than TNBC and non-TNBC tissues. It is well known that patients with TNBC have a worse prognosis compared to those with non-TNBC, primarily due to the lack of targeted therapies and the aggressive nature of the disease (*Carey et al., 2007*; *Yagata, Kajiura & Yamauchi, 2011*). In this study, we demonstrated that lower methylation levels of cg06268921 were significantly associated with poor OS and DFS in the TCGA-TNBC cohort. Although association was not validated in the external cohort, the discrepancy may be due to inter-cohort heterogeneity and limited sample size, as also reflected in the meta-analysis results. The meta-analysis revealed non-significant pooled effects with moderate heterogeneity ($I^2 > 60\%$), highlighting variability across datasets and the need for validation in larger, more homogeneous cohorts.
In addition to tissue-based assays, cfDNA in plasma is promising non-invasive diagnostic and prognostic markers for cancers. Some studies indicated that somatic mutation information in cfDNA can reflect intratumor heterogeneity and be used for differential diagnosis of BC (*Cailleux et al., 2022*; *Zhang et al., 2019*). However, the detection may be affected by clonal hematopoietic mutations and somatic mutations from non-breast tissues. In contrast, cfDNA methylation patterns are consistent with the cells or tissues from which they originated, making them a valuable alternative for cancer diagnosis (*Luo et al., 2021*; *Moss et al., 2018*; *Zhang et al., 2021b*). Several cfDNA-based methylation markers have been employed for BC diagnosis, yet few studies have explored their application in the differential diagnosis between the TNBC and non-TNBC (*Kloten et al., 2013*; *Liu et al., 2021*; *Zhang et al., 2021b*). Only one abstract from 2024 reported that a ctDNA-based targeted methylation sequencing assay correctly classified 84% (58/69) of TNBCs and 82% (94/115) of non-TNBCs (*Nance et al., 2024*). Despite its accuracy (82.6%, 152/184), sequencing remains expensive, time-consuming, and may require larger blood samples, which limited its widespread clinical application. In our cfDNA validation cohort, we successfully designed primers and probes for two candidate markers and developed a mddPCR assay with capable of simultaneously and independently quantifying the methylation level of each target gene. Meanwhile, compared to mqMSP, the mddPCR assay is more sensitive, with a 500-fold lower LOQ using only 10 ng template DNA. Furthermore, this method requires only 1–2 mL plasma, making it more suitable for routine applications and avoiding the high sequencing cost. We observed significant hypermethylation of cg06268921 in cfDNA from TNBC patients, consistent with methylation patterns seen in tumor tissues. Although cg23247845 methylation levels did not reach statistical significance, likely due to limited sample sizes, incorporating both markers into the cf-MDS, which could significantly distinguish TNBC from non-TNBC patients.

To further assess the specificity of our candidate methylation markers, we conducted a supplementary analysis using a public cfDNA methylation dataset tested by Infinium HumanMethylationEPIC BeadChip (GSE214344). This dataset includes plasma samples from five healthy individuals and seven luminal B BC patients. As shown in Fig. S11, seven of the eight tissue-derived markers showed no significant methylation differences between the two groups. Despite the small sample size and absence of TNBC cases, this finding aligns with our tissue-level results and suggests these methylation alterations may be TNBC-specific.

The early and accurate differential diagnosis of TNBC is crucial. Currently, immunohistochemistry is the primary method for determining BC molecular subtypes (*Bianchini et al., 2016*), but tissue biopsy and large mass detection may not be suitable for early-stage TNBC. Recent studies (*Shaikh & Rasheed, 2021*) have shown that non-invasive methods such as mammography and ultrasound, can also differentiate TNBC from non-TNBC. However, their discriminating power is relatively low (AUC = 0.719) and primarily focused on tumor morphological features (*Zhang et al., 2019*). Therefore, a new method is still needed to compensate for the above deficiencies. Our study confirmed the feasibility of cfDNA methylation markers for early-stage TNBC differential diagnosis. The cf-MDS

assay can differentiated stage I/II TNBC from non-TNBC in cfDNA cohort
(AUC = 0.701). While the diagnostic effect may be lower, combining cfDNA methylation
markers with existing methods could offer a more comprehensive and accurate approach
for differential diagnosis of TNBC in the future.

Several limitations of this study should be acknowledged. First, the sample size of the
plasma cfDNA validation cohort was not sufficiently large. The methylation-based panel
requires further validation in a multi-center study with a large-scale sample size. Second,
the candidate cfDNA methylation markers were selected from tumor tissues rather than
directly identified from plasma, which may limit the effectiveness in non-invasive
discrimination between TNBC and non-TNBC. Third, only two of the eight markers were
validated in plasma due to unsuccessful primer and probe design for the remaining six,
likely owing to unfavorable sequence characteristics. Future optimization or alternative
assay platforms may facilitate validation of these markers. Finally, we validated the
candidate DMCs using the TCGA dataset, which may cause us to miss some promising
markers, especially those not covered by the Illumina 450K array.

## CONCLUSIONS

In summary, our study identified and validated DNA methylation markers that show the
potential for the differential diagnosis and prognosis of TNBC, based on integration
analysis of in-house, TCGA and GEO methylation data. While our findings highlight the
accuracy of cfDNA methylation markers for detection TNBC, whether the markers can be
applied as non-invasive diagnostic and prognostic markers should be further validated in a
larger population.

## ACKNOWLEDGEMENTS

The authors thank the participants included in this study and their families.

### Funding
This work was supported by the grants from Heilongjiang Provincial Natural Science
Foundation of China (No. ZD2021H001), National Natural Science Foundation of China
(No. 82073643, No. 82073410), Nn10 Program of Harbin Medical University Cancer
Hospital (No. 2017-02), and Postgraduate Research and Practice Innovation Project (No.
YJSCX2023-27HYD). The funders had no role in study design, data collection and
analysis, decision to publish, or preparation of the manuscript.

### Grant Disclosures
The following grant information was disclosed by the authors:
Heilongjiang Provincial Natural Science Foundation of China: ZD2021H001.
National Natural Science Foundation of China: 82073643, 82073410.
Nn10 Program of Harbin Medical University Cancer Hospital: 2017-02.
Postgraduate Research and Practice Innovation Project: YJSCX2023-27HYD.

## Competing Interests

The authors declare that they have no competing interests.

## Author Contributions

- Lijing Gao conceived and designed the experiments, performed the experiments, analyzed the data, prepared figures and/or tables, authored or reviewed drafts of the article, and approved the final draft.
- Yanbing Li performed the experiments, prepared figures and/or tables, and approved the final draft.
- Chao Qu performed the experiments, prepared figures and/or tables, and approved the final draft.
- Yan Dong performed the experiments, prepared figures and/or tables, and approved the final draft.
- Qingzhen Fu analyzed the data, prepared figures and/or tables, and approved the final draft.
- Haibo Zhou performed the experiments, prepared figures and/or tables, and approved the final draft.
- Ning Zhao analyzed the data, prepared figures and/or tables, and approved the final draft.
- Xianyu Zhang conceived and designed the experiments, authored or reviewed drafts of the article, and approved the final draft.
- Da Pang conceived and designed the experiments, authored or reviewed drafts of the article, and approved the final draft.
- Yashuang Zhao conceived and designed the experiments, authored or reviewed drafts of the article, and approved the final draft.

## Human Ethics

The following information was supplied relating to ethical approvals (*i.e.*, approving body and any reference numbers):

The study was approved by the Medical Ethics Committee of Harbin Medical University (KY2016-01).

## Data Availability

Raw data is available in the Supplemental Files.

## Supplemental Information

Supplemental information for this article can be found online at http://dx.doi.org/10.7717/peerj.19888#supplemental-information.

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
