# Peer review of "Genome-wide discovery of circulating cell-free DNA methylation signatures for the differential diagnosis of triple-negative breast cancer"

_PeerJ, doi:10.7717/peerj.19888_

## Round 0.1 · original submission · Major Revisions

Reviewer 1 ·

Basic reporting

The authors performed a genome-wide analysis using an in-house Infinium HumanMethylationEPIC dataset, combined with TCGA and GEO data, identifying 113 differentially methylated CpGs. Then they used Lasso regression to further select eight optimal methylation markers. In cfDNA validation cohort, a mddPCR assay was developed to detect two of these markers (cg06268921 and cg23247845), achieving an AUC of 0.728 for differentiating TNBC from non-TNBC. The findings highlight the potential of methylation signatures as non-invasive diagnostic tools for TNBC. Overall, the manuscript is well-executed, though a few minor issues remain to be addressed. Here are my detailed comments.

Experimental design

1. In the methods section of the abstract, the authors mentioned using TCGA and GEO datasets to filter differential methylation sites (DMCs), but they do not provide specific screening criteria. It is recommended that the authors clarify the screening criteria to ensure that the selected DMCs are biologically meaningful and statistically significant.

Validity of the findings

1. The authors should clarify the rationale for changing the delta ß-value threshold from 0.1 for in-house analysis to 0.25 for the TCGA dataset.
2. The authors found that methylation level of cg06268921 is associated with overall survival (OS) of TNBC patients from TCGA. Disease-free survival (DFS) is also an important indicator to evaluate disease progression and survival. Thus, it is suggested that the authors further investigate the association between the methylation level of this site and DFS.

Additional comments

1.The manuscript shows inconsistency in the use of the “P” symbol (both lowercase “p” and uppercase “P” are used). I recommend standardizing the symbol to uppercase "P" throughout the text and figures for consistency.
2. Please provide the full definition of abbreviations upon their first appearance, such as “human epidermal growth factor receptor 2” for HER2.
3. Some of the cited references are relatively old; it is recommended to include more recent literature to support the manuscript.
4. Please include the full definitions of abbreviations in the figure legends as well.

Reviewer 2 ·

Basic reporting

Language is clear and appropriate.

Citations are mostly correct. Citations for glmnet and KEGG should be added.

The authors are interested in using methylation signatures of circulating DNA as a diagnostic for TNBC.
They rightly comment on the shortcomings on current diagnostic methods including IHC (invasive) and imaging techniques including mammography and ultrasounds (poor accuracy).
The authors point out how others have used cfDNA methylation patterns to discriminate between malignant and benign tumors with reasonable accuracy, and extend this idea to specifically identifying TNBC malignancies.

Figures and tables are legible and clear. Tables appropriately include patient metadata (Table 1) and information on the identified CpG sites.

Experimental design

The authors measured methylation patterns in tissues of TNBC and non-TNBC samples using a combination of their own samples and the TCGA publicly available dataset. Combining with the TNBC dataset, and removing

LASSO regression was used to perform feature selection with parameter shrinkage. This is an appropriate modeling technique when there are many strongly correlated features.

Validity of the findings

The authors identified eight markers with their model but unfortunately were only able to identify two in circulating plasma. This may be because they are simply not present in plasma, despite being useful markers readily observed in tissues. Were the authors able to identify any unmethylated DNA for these markers in circulating plasma? If not, this may further confirm the absence of these in plasma, methylated or not.

Adding a "normal" cohort of plasma would be an interesting further examination of these identified markers, and their degree of specificity to malignant tumors. While the authors are primarily interested in discriminating between TNBC and non-TNBC BRCA, the addition of healthy normal cfDNA from plasma would make for a more thorough analysis, and further explore the feasibility of using these markers (unmethylated and methylated) as a plasma-based diagnostic.

The validation cohort GSE69914 does include normal samples and would be worthwhile examining. Were the total of 8 markers observed in this validation cohort? It is not clear to me if Figure S8 and the validation cohort analysis made use of cg06268921, or in combination with cg23247845. If all 8 originally selected markers are present in the validation cohort, how does combined MDS score using all perform?.


Some of the non-TNBC patients are HER2+ with high proliferation scores. Do any of these have elevation of identified (cg06268921 and cg23247845) markers? It is a small dataset and further statistical analyses are not necessary or even feasible, but would be interesting for the reader.

An extended Figure S7-A subdivided further and labeled with HER2 and other status would be helpful. The fact that their markers were able to discriminate between lower grade TNBC and non-TNBC is already encouraging.

The authors leave open the possibility of further discriminating between low grade and high grade TNBC with cfDNA. Their failure to observe 6/8 identified markers from tissue data is a substantial hindrance, however.

Additional comments

add citations to glmnet, ChAMP, limma, pheatmap, KEGG, clusterProfiler, survival, survminer

Most important are glmnet (as is the basis of LASSO regression analysis and identification of 8 markers), and KEGG.


Figure S7A is not a histogram plot, but an ordinary barplot with separate samples on the x axis. Relabel for accuracy.

Reviewer 3 ·

Basic reporting

The authors reported that a total of 33 TNBC and 80 non-TNBC patients were enrolled in the study. However, genome-wide methylation analysis was conducted on only five TNBC and nine non-TNBC tissue samples. It remains unclear how the remaining patient samples were utilized or whether they were included in any other analyses within the scope of the study?

Experimental design

The authors reported that the DMCs were primarily enriched in signaling-related pathways (Fig. 2C) and in the biological process of protein binding (Fig. S1). However, the biological significance of these findings remains underexplored and should be addressed in the Discussion section to provide context on how these methylation changes may contribute to TNBC pathogenesis or progression.

Validity of the findings

Other studies in the literature have also identified differentially methylated regions (DMRs) in TNBCs (e.g., PMID: 36305646). It would be valuable to know whether the authors observed any of these previously reported DMRs in their analysis, or if their findings are unique to their specific cohort? If DMRs vary significantly between specific cohorts, it raises the question of how generalizable these findings are to the broader population of TNBC patients.

---

## Round 0.2 · accepted · Accept

The authors have addressed all of the reviewers' comments and manuscript is ready for publication.

Reviewer 2 ·

Basic reporting

Excellent

Experimental design

Much clearer.

Validity of the findings

Your additional analyses are very thorough and I believe make the work much more rigorous.

Reviewer 3 ·

Basic reporting

The authors have satisfactorily addressed the critiques raised during the review process. I have no further questions. The article can be accepted for publication.

Experimental design

-

Validity of the findings

-